# Housing Rabbit Does in a Combi System with Removable Walls: Effect on Behaviour and Reproductive Performance

**DOI:** 10.3390/ani9080528

**Published:** 2019-08-05

**Authors:** Alessandro Dal Bosco, Cecilia Mugnai, Melania Martino, Zsolt Szendrő, Simona Mattioli, Valentina Cambiotti, Alice Cartoni Mancinelli, Livia Moscati, Cesare Castellini

**Affiliations:** 1Department of Agricultural, Food and Environmental Science, University of Perugia, Via Borgo 20 Giugno, 74, 06100 Perugia, Italy; 2Department of Veterinary Science, University of Turin, Largo Braccini 2, 10095 Grugliasco (TO), Italy; 3Faculty of Agricultural and Environmental Sciences, Kaposvár University, 40, Guba S. str., H-7400 Kaposvár, Hungary; 4Istituto Zooprofilattico Sperimentale dell’Umbria e delle Marche, Via G. Salvemini 1, 06100 Perugia, Italy

**Keywords:** rabbit doe, group-housing, behaviour, welfare, reproductive performance

## Abstract

**Simple Summary:**

This study focuses on the welfare and reproductive performance of rabbit does housed in individual conventional cages (C) or in different colony cages: simple (does in the group for 100% of the reproductive cycle: C1) or combi (in both individual and group caging: C2). The results showed that C2 had some benefits compared to continuous grouphousing, but both colony systems achieved lower reproductive performance levels than the conventional system. Although C2 showed some improvement on the behaviour of does, the aggressiveness of group-housing the does to establish a rank order was responsible for injuries, higher disease risks, and higher kit mortality.

**Abstract:**

We evaluated the effects of two types of colony cages, in which rabbit does were always in a group (C1), and where they were in combi cages furnished with removable internal walls to allow both individual and grouphousing (C2), in addition to the control group (C: conventional individual cage), on welfare, reproductive performance, and global efficiency. Forty-eight New Zealand White nulliparous rabbit does underwent artificially insemination (AI) and were divided into three groups, and reared in the different systems for about 1 year. The reproductive rhythm provides AIs at weaning (30d). In the C1 system, does were continuously grouped, while in C2, walls were inserted four days before kindling and removed 1week after it (60% of the timesheet in group). Reproductive traits and behaviour were evaluated during the entire year. The behavioural observations were performed around days 7, 36, and 44, corresponding to the inclusion of the does in the maternal cages, the insertion of walls four days before kindling, and the removal of the walls 1week after parturition in the C2 group, respectively. The percentages of does with severe skin injuries and the distribution of the injuries on different parts of body were also registered. Does reared in conventional cages showed the greatest presence of stereotype behaviours, while the C1 group showed the highest (*p* < 0.05) incidence of aggressiveness after regrouping (attack, dominance features, and lower allo-grooming) in comparison to the C2 group (17% and 22%, in C2 and C1 does, respectively).Individually caged does achieved the best productive performance (sexual receptivity, fertility, kindling rate, and number of kits born alive and at weaning). The C1 group showed the lowest performance (*p* < 0.05), whereas C2 showed an intermediate one. Does housed in the combi cage (C2) had higher (*p* < 0.05) receptivity and fertility rates and higher numbers of kits born alive and at weaning (79.2% and 76.2%; 7.95 and 7.20, respectively) than the C1 group, but lower values (*p* < 0.05) than does that were individually housed.

## 1. Introduction

Domesticated rabbits originate from the European wild rabbit, which lives in territorial breeding groups consisting of an average of two to nine does, two to three bucks, and their progenies [1]. The rabbit is a social animal, and under natural conditions, lives in groups, establishing specific hierarchies within and outside the group. In particular, males compete for female access, while does compete for nesting sites. The social system of rabbits is not stable; in fact, usually, the males reach the group only in the mating period, which causes an increase in aggression and fighting. Moreover, wild rabbits show a particular maternal behaviour, consisting in only one visit a day at the nest-burrow by the mother, with about 3min of suckling. This behaviour is related to the protection of the kits from predators.

Thus, when rabbits were first domesticated, they were reared in groups. The origin of the *Leporia* dates back to the Middle Ages following the delimitation of areas destined for hunting [2]. Due to several behavioural and hygienic problems, and consequently, poor productivity, housing rabbit does in groups was abandoned in France in the late 1970s [3]. Numerous advantages—for instance, the introduction of wire-mesh cages, intensively selected genotypes, artificial insemination, cycled reproduction, balanced pelleted feed, and automatic feeders—were important steps towards intensive rabbit production [2]. However, most of the infrastructure used in intensive systems requires that the rabbits are reared in single cages without any or low social interaction, not meeting the ethological needs of the animals.

Current research shows that the continuous group housing of does, independent of the larger area for moving and social contact, contradicts the welfare standards, often resulting in chronic stress, aggressiveness and injuries, as well as a higher risk of disease and mortality [4]. At the same time, the reproductive performance of group-housed rabbits is lower, and the production costs are higher when compared with individual housing systems.

Therefore, semi-group housing with defined consecutive periods of single and grouphousing has been proposed [5,6,7,8], albeit without solving the problem of aggressiveness linked to the regrouping of does. The authors concluded that in the near future, the use of group-housing systems for does with kits does not seem to be a realistic practice, because of numerous unsolved welfare problems. Hoy and Matics [9] drew the same conclusions; in particular, they affirmed that in the near future, the single housing of does with kits with some enrichment would be the common housing system in intensive rabbit production. Szendrő et al. [10], summarizing the advantages and disadvantages of the individual, combined, or grouphousing of does, proposed individual housing, but underlined the necessity of environmental enrichments. Thus, the colony cages proposed for does do not meet the needs of the animals and negatively affect the well-being and the reproductive performance of rabbit does (pseudopregnancy, abortion), and should therefore be modified to be more efficient. In January 2017, members of the European Parliament’s Agriculture Committee voted in favour of a report that set out key improvements for rabbit welfare (growing rabbits and does), and consequently, other studies to solve the welfare problem of housing systems are needed, with the aim to develop a single-cage module for mother and kits without any external needs.

Based on the above-mentioned considerations, and regarding the management difficulties of the colony cage, the aim of this study was to evaluate the welfare and reproductive performance of rabbit does for meat production, reared in either two different colony cages or in commercial individual cages. To our knowledge, this is the first experiment where different colony cage systems, one consisting of continuously grouped does and the other of individually or grouped (for 60% of the reproductive cycle) does are compared.

## 2. Materials and Methods

### 2.1. Animals and Housing

The research was carried over the course of 1 year at the experimental rabbit farm of the Department of Agricultural, Food, and Environmental Science of Perugia University (Italy). The experimental design was performed following the EU Directive 2010/63/EU [11] for the protection of animals used for scientific purpose.

All the considered experimental groups were positioned in the same artificially ventilated (0.3 m/s) [12] building; environmental temperature and relative humidity were daily controlled (range: +15/+28 °C; 60%/75%, respectively), and the lighting schedule was 16L/8D. 

Two different colony cages were developed in collaboration with Metac–Ellebis.r.l. manufacturing (Fabriano, Italy):-Simple colony cage (C1; Figure 1), with dimensions of 76W × 150L × 60H cm, and four external shut-out nest boxes (38 × 25 × 35 cm) at each side of the cage.-Combination colony cage (C2; Figure 2) with two levels: in the lower level, the maternity cage was located with eight units and nests, divided into two sides (four rabbit does per side), furnished with removable walls to manage group and individual housing, depending on the different production phases. The dimensions of cage were of 130W × 158L × 60H cm, with eight 39.52 × 20 × 35 cm external shut-out nest boxes at each side of the cage. In the upper level, there was a multi-purpose cage with removable walls, available for hosting the non-pregnant and severely injured does and for potential sopra-numerous pregnant does (possibility to add nests). This cage was considered an autonomous and independent production unit. -Conventional cage (C), with dimensions of 38W × 60L × 34H cm, provided with an external nest box (38 × 25 × 35 cm). 

In the colony cages (both C1 and C2), rabbit does were housed at the same density as in the conventional system (C). 

### 2.2. Experimental Design

To evaluate the effect of cage type (C vs. C1 vs. C2) and housing conditions (total isolation vs. grouphousing vs. semi-grouphousing) on the welfare and reproductive performance of rabbit does, all cages were wire-mesh barren, and were neither enriched nor provided with bedding materials.

Forty-eight homogenous nulliparous New Zealand White rabbit does 120 days old were purchased by the Italian National Rabbit Genetic Reference Center (Foggia, Italy), underwent artificially insemination (AI), and were divided into the three groups and randomly placed into one of the housing systems, as follows: -C1 group (*n* = 16, four does × four production units–open system);-C2 group (*n* = 16, four does per block × four production units–closed system);-C group (*n* = 16, one doe × 16 production units).

In all experimental groups, non-pregnant does at any cycle of AI were replaced for those having four contemporary kindling. 

In particular, after abdominal palpation, executed at 12 d post AI, the non-pregnant does were replaced with pregnant ones hosted in the upper level of the cage (C2), or in supplemental single cages (C and C1). Considering the average fertility rate of the three experimental groups, about 1 doe/cycle was replaced because of being not pregnant. 

During the first 16 days of lactation, controlled nursing was performed by permitting the does access to the nest only once a day for 15 min. Milk output was determined by weighing the doe immediately before and after suckling [12].

All does were inseminated immediately after weaning (30 d) for about 1 year, with six consecutive reproductive cycles. The AI was performed in the morning by inoculating 0.5 mL of diluted fresh semen, containing about 10 million spermatozoa [13]. No oestrus synchronisation was applied. Ovulation was induced by intravaginally inoculating 10 μg of GnRH (Receptal^®^ Intervet International Boxmeer, Holland). After abdominal palpation, executed at 12 d post-AI, the non-pregnant does were replaced with pregnant does hosted in the upper level of the cage (C2) or in supplemental single cages (C and C1). 

Does housed in C1always remained in the group, and after kindling, they remained with their kits until weaning (30 d). Subsequently, the kits were moved into bicellular cages, and the does were re-inseminated [10] and remained in the same cage until the next kindling (Figure 3).

The C2 does remained in the group until four days before parturition, when the internal walls were installed to permit individual housing, in order to avoid disturbance by other females during kindling. One week after parturition, the internal walls were removed, and the four does were grouped again with their litters until weaning (30 days). After weaning, the kits stayed in the same colony as the fattening cage until 70 days of age, whereas the does were moved to the other side of the cage, where they were artificially inseminated and housed in the colony. Then, the next productive cycle started until inserting the walls, four days before kindling (Figure 4).

In the upper cages, four does per production unit were placed to replace the non-pregnant does at every insemination cycle (dotted lines). 

After the fattening cycle (70 days), growing rabbits were transported to the slaughterhouse, and their cages were cleaned, disinfected, and prepared for housing the rabbit does coming from the other side of the cage.

### 2.3. Diets

A commercial diet with the following chemical composition was distributed once a day manually *ad libitum*: crude protein, 18.7%; ether extract, 4.8%; crude fibre, 14.7%; ash, 9.2%; NDF,29.2%; ADF, 18.5%; ADL, 3.3%; cellulose, 14.5%; hemicelluloses, 10.6%; and 10.9 MJ/kg digestible energy. The cages were equipped with hopper feeders and drinkers.

### 2.4. Behaviour Observation and Ethogram

For each colony group (C1 and C2), eight does were randomly chosen and marked with spray can of different colours on their backs for identification; for group C, cage identity was attributed to the observed doe (*n* = 8). To develop the ethogram, behaviour patterns and categories of doe behaviour (Table 1) were observed for three consecutive days before starting the trial; on the basis of those recorded, behaviours was designed the table.

In the colony cages, the following social relationships were recorded: smelling the others, allo-grooming, attack, and dominance and submissiveness features. A doe was considered to be dominant when mounting, biting, and scratching another doe was observed, or when she was sitting with a tense body posture with erected ears and tail near to another doe (dominance features), instead of performing a crouched posture avoiding visual contact, rolling over on the back, ears back, and tail tucked (submissive features) [14].

The rabbits were observed during daytime, between 9–12 a.m. and 2–5 p.m.; thus, the activity during night was not recorded. The recording of behaviours was performed daily for seven consecutive days, during each of the six reproductive cycles, in the following periods:from days 1 to 7, when does were in the colony (C1 and C2);from days 30 to 36, around the kindling period, when in group C2, the internal walls were present and animals were housed individually. The behavioural observations were interrupted on the day of kindling to provide a peaceful and quiet environment for does. In C1, the does were always in the colony;from days 38 to 44, when in group C2, the internal walls were removed and the does were grouped again.

The behaviour patterns were recorded by two trained operators in the morning (9–12 a.m.) and in the afternoon (2–5 p.m.), using the focal animal sampling method [15]. Prior to each observation period, the operators waited 5 min to permit the animals to adapt to their presence.

To establish the end of a performed behaviour, 5 s were allowed to determine if the same behaviour was repeated; after this time, a new behaviour pattern started [15]. For each doe, the number of times a particular behaviour occurred, with reference to total observations, was converted into a percentage. Each behaviour of an individual doe was added together and divided by 8 to give a mean percentage for each observation period. 

In the colony groups, to permit the recording of the social relationships, observation was extended by 3 min. Patterns of social behaviours were analysed separately, calculating their frequency as a percentage of total social relationships. The does of group C were observed for a daily period of 48 min (8 does × 3 min each = 24 min in the morning, as well as24 min in the afternoon), whileC1 and C2 does were observed for a daily period of 96 min (8 does × 6 min each = 48 min in the morning and 48 min in the afternoon).

### 2.5. Body Injuries

Skin injuries were scored as an indicator of aggression among all does in the colony groups. The presence and gravity of skin injuries on does were assessed in the periods of the behavioural observation. Different parts of the body (body and limbs, head and ears, genitals, tails) were analysed according to Kalle [15], using the following scale: 0 = no injuries; 1 = superficial bites (<1 cm) that normally heal within a couple of days; 2 = moderate to severe injuries (>1 cm); and 3 = open wounds.

### 2.6. Reproductive Performance

The following reproductive traits were recorded: sexual receptivity (at AI, colour and turgescence of the vulva: a doe was judged receptive when her vulva was red or purple and turgid), fertility and kindling rate (pregnant/AI × 100 and kindling/AI × 100, respectively) and number of kits born alive and at weaning. As previously indicated, fertility rate was established by abdominal palpation at 12 days after AI. Late fetal mortality was estimated as the difference between fertility and kindling rates. After three consecutive AIs, does that were never pregnant were replaced by does of the same age.

The body weight of does was measured at kindling and at weaning; kits were individually weighed at weaning. Kit mortality was recorded daily. The indices of efficiency were calculated in terms of overall productivity (number and weight of rabbits sold/year/doe) and production losses (difference in kg between theoretical production, considering fertility rate = 100%, mortality of the young rabbits = 0%, and kindling interval = 60 days, and the actual rate) [16].

On the day of AI, all does were submitted to ultrasound scanning (ALOKA model SSD-500,) at the perirenal regions (3 cm ahead of the second–third lumbar vertebrae), after shaving that part with a hair trimmer. Scapular fat thickness was measured directly, and the average of two measurements (left and right side) was calculated. Perirenal fat depots were estimated using a regression curve, as described previously [17].

### 2.7. Statistical Analysis

Statistical analysis of behaviour patterns was performed using a mixed linear model [18], considering the fixed effects of the housing system and the productive phase, the random effect of does, and their interaction. The significance of differences was evaluated by the *t*-test (*p* < 0.05). Preliminary analysis showed no differences in the behavioural observations in the morning and in the afternoon; thus, these sub-periods were pooled. Reproductive performances were analysed with a linear model comprising the fixed effect of housing system. Non-parametric variables (sexual receptivity, fertility, pre-weaning mortality, and annual replacement of does) were analysed with the Chi-square test.

## 3. Result and Discussion

### 3.1. Behaviour and Welfare of Does

Most of the behavioural patterns of does were strongly affected by the housing systems (Table 2). Buijs et al. [7] show that rabbits spend a great percentage of daytime hours stationary, while night-time hours are spent grooming and ingesting. Similarly, according to Jilge and Hudson [19], static activities are the most common behaviours in all groups of does during the day.

Feeding, drinking, defecation, and urination, which are the main basic life functions, were independent of the housing system. The largest differences were found between individual housing and colony groups. Higher (*p* < 0.05) frequencies in jumping, lying down, and standing up were observed in the two colony groups (C1 and C2), while the frequencies of stereotypical behaviour and crouching were higher in individually housed does. The moving behaviour was strongly influenced by system and time, with higher frequencies in the colony systems; in the C2 system, as expected, during the isolation period, the frequency of movement strongly declined to values comparable to those of the single cage.

These results are in agreement with the findings of Lawrence and Rushen [20], who observed some abnormal behaviours, such as smelling, licking, and biting bars in animals housed in single cages. According to Verga et al. [21], stereotypes are expressed more in individually housed does.

Generally, no significant differences were observed in behaviour patterns between the two colony groups, except moving, which was higher in C1 than in C2 rabbits (30–36 days), whereas smelling bars, lying down (30–36 days), allo-grooming, and dominance features (38–44 days) were higher in the C2 group.

In our previous study [22], we assessed the behavioural patterns of does housed in C1 colony cages that were trained (putting the same doe in the same nest and holding it inside for 10 min during the first two days in the colony cage) or not trained to go into their own nest, but the production did not reach the results of does housed in single cages. Accordingly, in the present study, the reproductive performance of C does, which had no social relationships, was higher in terms of receptivity and fertility rate, as well as in the number of live-born kits. The C1 group showed the lowestresult for both reproductive and productivity traits, probably due to the higher social pressure and aggressiveness, as confirmed by the percentage of does injured and replaced; this is in accordance with ourprevious study, where the untrained does showed the same trend. Compared to the C1 group, the C2 does reached better indices of global productivity (higher number of rabbits sold/year/doe and live weight sold/year/doe, as well as lower production losses, kindling intervals, and annual replacement of does), and can be compared with the trained does, thanks to the internal walls that allowed a more comfortable environment during the critical phase of kindling. According to these findings, the reproductive performance of C2 females that did not have any social relationships along the peri-partum period were higher in terms of receptivity and fertility rate, as well as the numbers of live-born kits and milk production when compared toC1 females, whereas such parameters had lower values when compared to C does.

The maternal behaviour of does continually maintained in the colony (C1) consisted of the entry of some does into the nest box of other does during the peri-partum period, and some aggression of the dominant doe. Naturally, C2 does, being restricted, did not show this undesirable phenomenon.

Other behavioural patterns were affected by the phases of the reproductive cycle (Table 2). The frequency of moving decreased and smelling bars increased in the C2 group when the internal walls were inserted (30–36 days) and the does stayed alone.

At days 38–44, when the does were in the group with the litter, moving increased and smelling bars decreased. The opposite trend was observed in moving and smelling, corresponding to when the rabbits were in the cage alone or in the colony. Individually caged rabbits can move less than rabbits in the colony cage; at the same time, the stereotypical behaviours (e.g., smelling and biting bars) could be more frequent [23,24].

In C1 does, the frequency of laying down and dominance were lower at days 30–36 than in the other periods. Of course, social contacts were observed only in the colony groups.

In the present study, the attacks among does were few (≤3.12%), and no differences were found in the frequency of attacks between the colony groups (C1 vs. C2) and periods. Although the timing of the establishment of a hierarchy in a group of does could be different, in the present experiment, it was relatively short (≤24 h from grouping/regrouping), and the frequency of attacks was low and stable during the different productive phases. It may be assumed that the composition of the groups was more uniform than in other experiments.

According to several authors [7,25,26,27,28,29,30], does reared in groups (stable or temporary) show a high frequency of aggressiveness before establishing the hierarchies at the grouping/regrouping of the rabbit does. Zomeño et al. [31] also reported an extremely short period of aggressiveness (biting, boxing, chasing, ripping, carousel fights, threatening, and attacking) after group formation, with a large variation among pens.

The percentage of injured does (head and ears, body, tail) was unchanged in the C1 group (Table 3), suggesting that once a hierarchy was established and not disturbed, it was maintained for a long time. Injuries were often found on the tail in all groups with a major score of 2 (moderate), for which we cannot offer any explanation.

In the C2 group, when the walls between the cages were moved and the does were in groups again (days 38–44), the percentage of injured rabbits increased, as did the gravity of those injuries, mainly belonging to score 2(moderate to severe). Every time the walls in the C2 cages were removed, the stability and the social hierarchy of the groups was broken, and fighting for a better position in the dominance order appeared again. This situation is similar to what has been reported by other authors when regrouping does [27,28,29,30,31,32]. It should be noted that in C1 and C2, at every reproduction cycle, one non-pregnant doe of the original group could be replaced by another pregnant one to establish the expected number of kindlings (*n* = 4).

Aggressive behaviour is also widely observed in wild animal species [33], mainly in group-living species. A dominance hierarchy exists among the females and separately between the males of European wild rabbits [33,34,35]. At the start of the reproductive season, the fights are intense; then, when the dominance hierarchy is established, aggressive behaviour and fighting occur less often during the entire reproductive season [35,36]. After parturition, the does stay near to their nest and are intolerant against other rabbit does [36]. Albonetti and Farabollini [37] also found a great reduction of the aggression frequency after the establishment of a hierarchy in rats, and suggest that social interactions between animals are friendly.

According to Andrist et al. [27], on Swiss farms, there are agonistic interactions, with lesion rates of 28% and 40%without and with regrouping, respectively.

Accordingly, the main behavior problems in C1 and C2 appeared after the regrouping of does, when aggressiveness, fighting, and injured rabbits are frequently observed [24,29,30,38]. Different environmental enrichments and strategies (platform, PVC pipe, hiding place, straw, territory, dark corridors, group stability, regrouped into home or new pen, and sprayed odours) were tested to reduce the high percentage (40%–60%) of injured rabbit does, albeit without success [7,28,30,31,32,33,34]. However, the aggressive interactions decrease some hours/days shortly after group formation [24,29,30,33,39].

In conclusion, from the behaviour point of view, it can be assumed that individual caging during peri-partum avoids the exchange of nest negatives for the litter, but does not eliminate fighting for the rank order.

It seems that some other methods should be investigated to reduce aggressive behaviour, fighting, and injuries after grouping the does. In particular, other periods of grouping/regrouping, possibly by reducing the amount of time spent in the group, should be studied.

### 3.2. Reproductive Performance of Does

The reproductive performance of single and colony-housed does are shown in Table 4. Although the extensive reproductive rhythm should improve the condition and consequently, the reproductive efficiency of does [40], both colony groups showed a lower fat depot (32 and 40 g/doe, respectively, in C1 and C2) than the control does (53 g/doe), and did not reach a similar performance.

It has been reported that in rabbit does, suckling determines a high fat mobilization [39,41] and energy deficit, possibly causing hypo-fertility [17].

The C1 group showed the worst results, probably due to the high social pressure and aggressiveness, as confirmed by the percentage of does injured, the high annual replacement, and the numerous mating attempts between females, which probably caused pseudo-pregnancy [4,24]; this is in accordance with a previous study, where does untrained for grouphousing showed the same trends [42].

Other authors have confirmed that rabbit does that are continuously together show decreased reproductive performance because of stress and pseudo-pregnancy [30]. Kit mortality is also high because of competition among does for nests and aggression towards kits [8,24,28,30,43,44,45]. Moreover, a high level of embryo mortality, calculated from the difference between fertility and kindling rate, was shown in the C2 group and, to a higher extent, in the C1 group (6.2% and 9.1%, respectively, vs. 4.5% in C).

C2 does that did not have any social relationships around the peri-partum period reached higher receptivity, fertility, and kindling rates; in addition, the numbers of kits born alive and at weaning were higher when compared to the C1 group, while these parameters were lower than in the C group. The reproductive performance of individually housed does, which did not have any direct social relationships (e.g., fighting, stress), was characterized by the highest receptivity, fertility, and kindling rates, as well as the highest number of kits born alive and weaned.

Because at kindling and in the first part of the lactation period, rabbit does of the C2 group were housed individually, their reproductive performance was better than that of rabbit does that were continuously reared in a group. However, their production was lower than that of individually housed rabbits, because they still experienced social stress, higher levels of aggressiveness, and more frequent body injuries.

Maertens et al. [44,45,46,47,48] reported reproductive performance levels in part-time grouped does (similar to the C2 group) comparable with those of does reared in individual cages. However, in these experiments, the housing system was equipped with an elevated platform, and the does were re-grouped later (18 days after parturition).

It is assumed that stress induces an increase in plasma prolactin [49,50], which is responsible for the antagonism with gonadotrophin hormones [51]. Bench and Gonyou [50] indicate that stress can reduce fertility by affecting the frequency and amplitude of luteinising hormone (LH) pulses, ultimately depriving the ovarian follicle of adequate LH support. This will lead to reduced oestradiol production by slower-growing follicles. Rommers et al. [30] and Theau-Clement et al. [51], studying group-housed does, attribute the low reproductive performance to pseudo-pregnancy.

In our study, the aggressiveness of doe relationships and the presumable incidence of pseudo-pregnancy and embryo mortality might have contributed to the lower reproductive performance, especially in the C1 group. The stress observed in this group might have triggered maternal failure, resulting in nest-building failures and low kit survival. The entry of does to the nest box of others and the aggression of the dominant doe [24], observed in the C1 group, were responsible for the low kit survival rates and the high percentage of injured does.

There is a multiplicity of hormones and cerebral structures involved in the maternal behaviour of does, particularly during nest-building [52,53]; stress acts upon neuro-hormonal centres, leading to alterations affecting hormone release. Moreover, the behaviour patterns of both a mother and her kits during peri-partum and lactation are stereotypical and lack flexibility, which is characteristic of most mammals [54]. This suggests that any alteration of the environment in which birth and nursing take place, e.g., stressful episodes, could lead to a failure in the ability of kits to survive, because they entirely depend on maternal care (nest building, nursing, suckling).

Regarding the global productivity (Table 5), the C2 does showed about 15% lower values than the does from individual cages. The high annual replacement was mainly due to bodily injuries. On the other hand, it should be underlined that C2 does, in comparison to C does, showed social interactions and higher locomotor activity.

One of the most important factors for the reproductive success of group-housed females could be the social rank: subdominant does suffer from high stress [1,22], and the higher reproductive success of the dominant does was probably caused by their better physical condition. They probably have higher body weights, lower corticosterone levels, and lower heart rates than subdominant females [38]; as the immune system is highly correlated to the social position of the animal, social rank may be a mediator of diseases [54].

## 4. Conclusions

Some problems of the group-housing system (C1), where does are continuously together (e.g., pseudo-pregnancy, double littering, high embryo mortality), can be solved using the proposed housing system (C2). This combined system for does and fattening rabbits could be a viable strategy in the future, mainly because it combines the benefits from the hygienic and welfare points of view, as well as high production levels.

However, there are still some negative aspects to solve, such as aggressiveness (e.g., injuries increased, which is in contrast with animal welfare), lower productivity than that of the traditional single-cage system, and higher production costs. Against this background, it would be interesting to investigate the housing system studied (C2) with the addition of cage enrichment to try to lower aggression levels when making changes within the rabbit group.

At present, continuous grouphousing of rabbit does is neither good for the rabbits (welfare) nor for the farmers (performance). Therefore, before proposing different grouphousing systems for rabbit does, a significant amount of research is needed.

## Figures and Tables

**Figure 1 animals-09-00528-f001:**
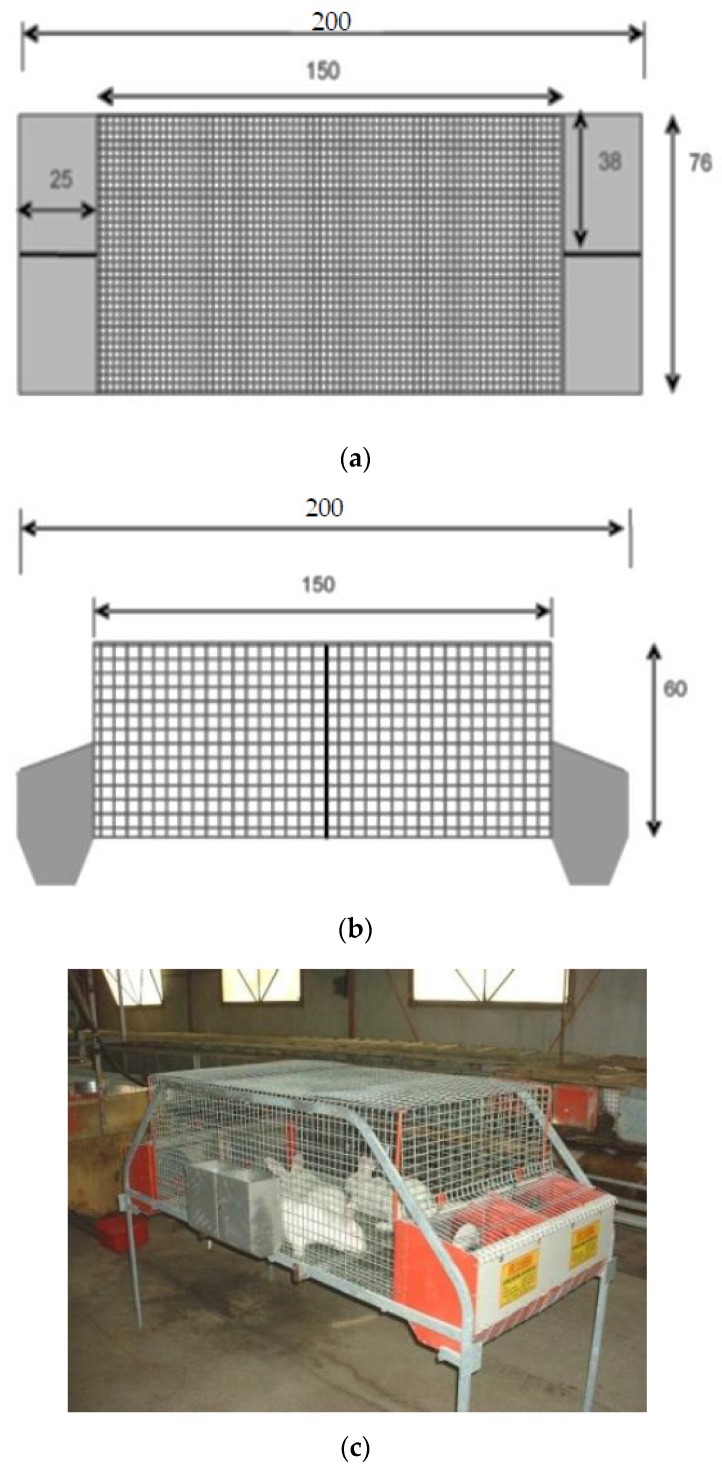
Dimensions (cm) of continuously grouped (C1) colony cage: (**a**) front view, (**b**) side view, (**c**) lateral view.

**Figure 2 animals-09-00528-f002:**
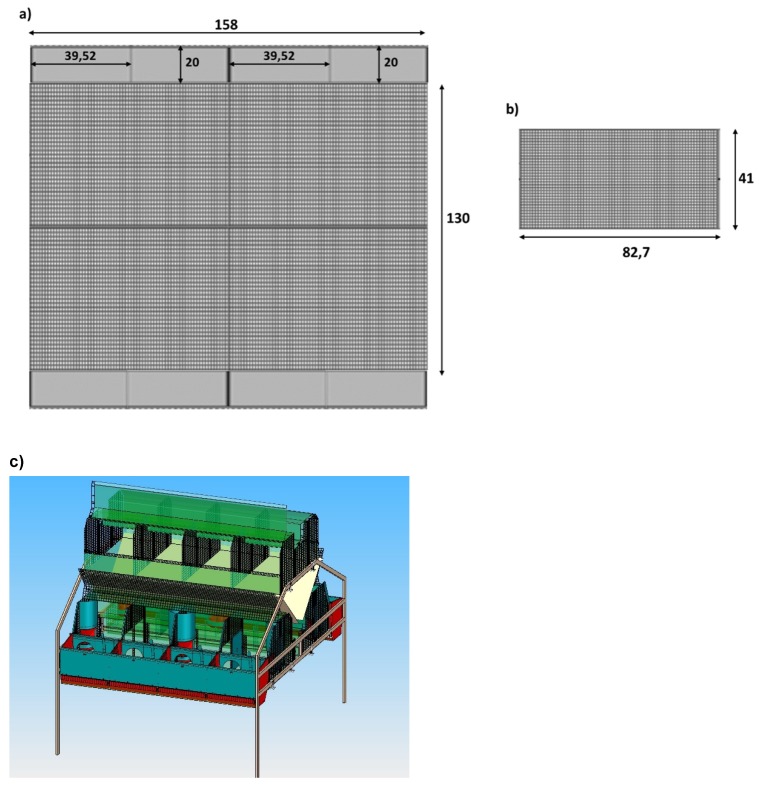
Dimensions (cm) of the combination(C2) colony cage: (**a**) lower level, (**b**) upper level, and (**c**) lateral view with the particulars of removable walls.

**Figure 3 animals-09-00528-f003:**
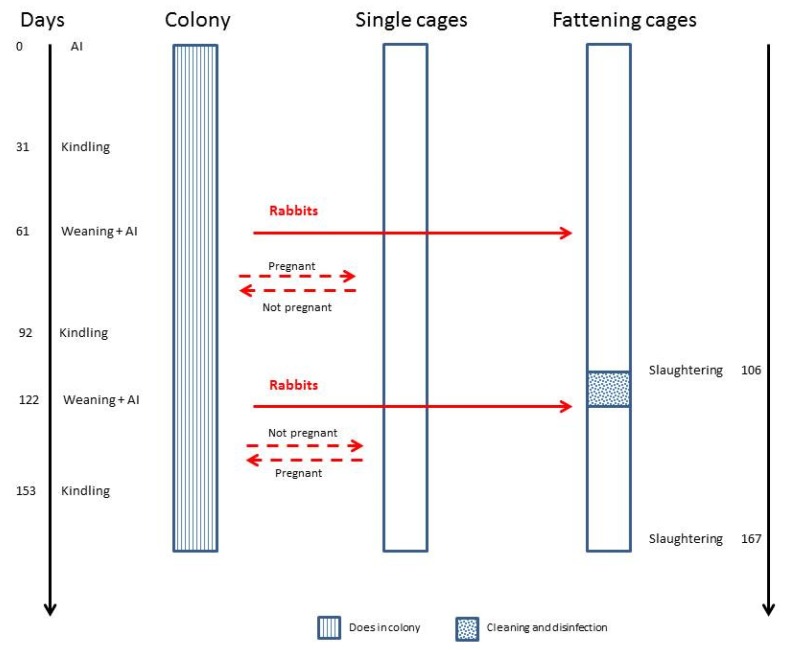
Scheme of rabbit production protocol of C1 cages. The production cycle is open and requires “external” cages for replacing the non-pregnant does and for housing the weaned rabbits until slaughtering.

**Figure 4 animals-09-00528-f004:**
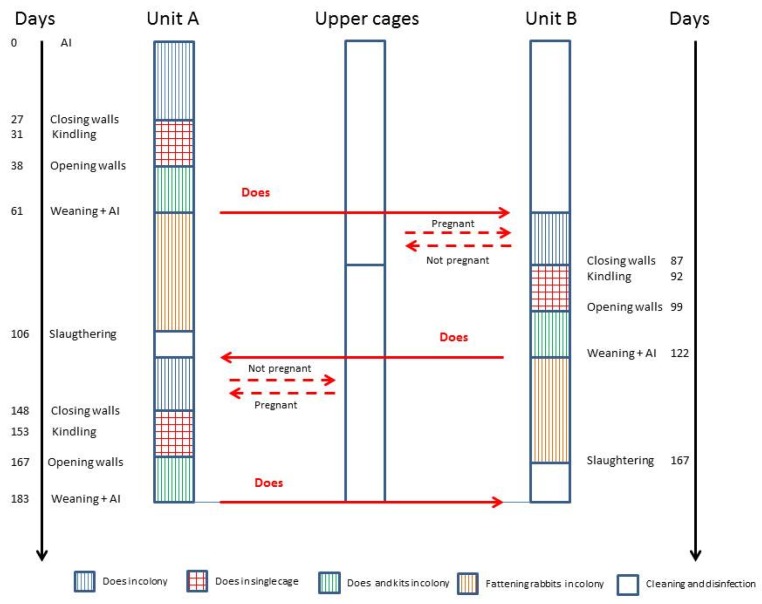
Scheme of rabbit production protocol of C2 cages. The production cycle is closed: non-pregnant does were housed in the upper cages, and young rabbits after weaning were kept in unit A or B, alternatively.

**Table 1 animals-09-00528-t001:** Does’ ethogram, categories of behavior patterns, and behavior description.

Categories	Behavior Patterns	Behavior Description
Move	Moving	Any movement in any direction where all four limbs are involved
	Jumping	Voluntary movements of jumping
Eat	Eating	Head above the feeder. Eating or chewing pellets
	Drinking	Head in close proximity to water nipple. Nosing or drinking from water nipple
Self	Comfort	Licking, scratching, or nibbling of the body
Stereotypies	Biting bars	Licking or gnawing cage bars and scratching cage floor insistently
	Smelling bars	Smelling bars and cage floor insistently
Static	Lying down	Resting with chest or stomach on the floor. Fore limbs stretched in front of the body
	Crouching	Resting with chest or stomach on the floor. Hind and fore limbs crouched under body
	Sitting-up	Sat in upright position on hind limbs and fore limbs straight, but without bust touching the floor
	Staying	Standing still on four straight limbs
Standing alert	Standing up on the hind legs	Sitting in upright position with ears erect
	Standing up on hind legs with erect ears	Sitting in upright position on hind limbs and fore limbs straight, with ears erect
Maternal	Nesting	Nest-building consists of digging a burrow, collecting straw, and shaping it into a nest inside the burrow, as well as plucking body hair and lining the straw nest with it
	Change of nest	A doe that enter in another doe nest
Others	Defecation, urination caecotrophy	-
Social relationship	Smelling other	Smelling another doe
	Allo-grooming	Licking, scratching, or nibbling another doe’s body
	Attack	Offensive moves, in which the doe attempts to bite its opponent
	Dominance feature	A doe that mounts, bites, or scratches another doe, or that sits with a tense body posture with erected ears and tail near to another doe
	Submissive feature	A doe in a crouched posture that avoids visual contact, rolls over onto the back, ears back, and tail tucked (submissive features) near to another doe

**Table 2 animals-09-00528-t002:** Effect of housing system on percentage of behaviours (with respect to total activities) and percentage of social behaviours (with respect to total social activities) in different periods of the reproductive cycle.

Period		1–7 Days			30–36 Days			38–44 Days			*p*-Value			MSE
Behavior Patterns	Categories	C	C1	C2	C	C1	C2	C	C1	C2	*T*	*S*	*T* × *S*	
Moving	Move	7.33 ^a,b^	9.80 ^b^	9.60 ^b^	4.01 ^a^	10.70 ^b^	2.69 ^a^	4.08 ^a^	9.62 ^b^	9.70 ^b^	*	*	*	2.95
Jumping		0.57 ^a^	1.80 ^b^	1.88 ^b^	0.92 ^a^	1.80 ^b^	1.77 ^b^	0.11 ^a^	1.69 ^b^	1.29 ^b^	n.s.	*	n.s	0.18
Eating	Eat	8.29	9.50	9.88	10.74	10.66	10.70	11.00	8.67	9.91	n.s.	n.s.	*	1.09
Drinking		3.55	2.70	3.20	3.00	3.89	2.75	11.18	2.61	3.49	n.s.	n.s.	n.s	0.88
Comfort	Self	8.65	9.70	9.90	10.93	8.23	8.34	3.24	8.68	9.15	n.s.	n.s.	ns.	1.75
Biting bars	Stereotypies	8.71 ^b^	2.04 ^a^	2.64 ^a^	6.43 ^b^	2.77 ^a^	2.53 ^a^	20.04 ^b^	2.76 ^a^	2.25 ^a^	n.s.	***	**	1.48
Smelling bars		7.44 ^b^	3.91 ^a^	3.83 ^a^	7.66 ^b^	3.65 ^a^	5.62 ^a,b^	8.44 ^b^	3.53 ^a^	3.14 ^a^	*	**	*	1.20
Lying down	Static	7.7 ^a^	15.89 ^c^	15.70 ^c^	9.50 ^a,b^	13.00 ^b^	16.17 ^c^	3.90 ^a^	15.76 ^c^	16.30 ^c^	*	**	*	1.55
Crouching		28.15 ^b^	10.32 ^a^	10.61 ^a^	32.15 ^b^	10.20 ^a^	17.32 ^a^	28.91 ^b^	10.43 ^a^	10.33 ^a^	*	**	**	2.07
Sitting-up		3.56 ^a,b^	6.30 ^b^	5.74 ^b^	1.17 ^a^	5.65 ^b^	5.44 ^b^	1.01 ^a^	6.40 ^b^	6.62 ^b^	*	**	*	0.74
Staying		7.78 ^b^	4.84 ^a,b^	4.26 ^a,b^	4.94 ^a,b^	3.76 ^a,b^	5.50 ^a,b^	1.06 ^a^	4.74 ^a,b^	4.24 ^a,b^	n.s.	*	*	1.11
Standingup on the hind legs	Standing alert	5.00 ^b^	5.56 ^b^	5.58 ^b^	0.53 ^a^	5.50 ^b^	6.41 ^b^	0.00 ^a^	5.86 ^b^	5.47 ^b^	*	**	*	1.05
Standingup on hind legs with erect ears		1.68 ^a^	8.50 ^b^	7.57 ^b^	1.48 ^a^	8.3 ^b^	8.65 ^b^	1.00 ^a^	8.8 ^b^	8.4 ^b^	n.s.	***	n.s	1.58
Nesting	Maternal	-	-	-	5.74	3.11	4.60	-	-	-	-	-	-	0.82
Change of nest		-	-	-	-	8.50	-	-	-	-	-	-	-	0.15
Defecation, urination caecotrophy	Others	1.59	1.12	1.34	0.80	1.43	1.56	0.70	1.25	1.30	n.s.	n.s.	n.s.	0.25
Smelling other	Social relationship	-	2.80	2.80	-	1.12	-	-	3.37	2.22	-	-	-	0.61
Allo-grooming		-	1.46 ^a,b^	1.66 ^a,b^	-	1.01 ^a^	-	-	0.83 ^a^	1.98 ^b^	-	-	-	0.47
Attack		-	2.60 ^b^	2.87 ^b^	-	3.12 ^c^	-	-	2.00 ^a^	2.70 ^b^	-	-	-	0.52
Dominance feature		-	0.68 ^a^	0.60 ^a^	-	1.78 ^b^	-	-	1.50 ^b^	1.95 ^c^	-	-	-	0.41
Submissive feature		-	0.48	0.34	-	0.39	-	-	0.54	0.67	-	-	-	0.15

*n* = 144 (8 does/C group + 8 does/C1 group + 8 does/C2 group) × 6 breeding cycles]. C: conventional housing, C1: group housing, C2: semi-group housing. *T*: period effect, *S*: system effect. Different letters on the same rows^(a–c)^ mean different *p* values. * = *p* ≤ 0.05, ** = *p* < 0.01, *** = *p* < 0.001. n.s. = *p* > 0.05.

**Table 3 animals-09-00528-t003:** Effect of housing system on skin injuries in different periods of the reproductive cycle.

Period	Body Part	1–7 Days		30–36 Days		38–44 Days		*p*-Value			*X* ^2^
Housing system		C1	C2	C1	C2	C1	C2	*T*	*S*	*T* × *S*	
Injured animals %		19 ^a,b^	20 ^b^	24 ^b^	-	17 ^a^	22 ^b^	*	**	n.s.	2
Part of body	Head and ears	24 ^a^	23 ^a^	25 ^a^	-	26 ^a^	36 ^b^	*	**	*	4
	Body	10 ^b^	10 ^a^	7 ^a^	-	7 ^a^	10 ^b^	*	*	*	2
	Genitals	-	-	-	-	-	-	-	-	-	-
	Tail	66 ^b^	67 ^b^	68 ^b^	-	67 ^b^	54 ^a^	*	**	*	5

*n* = 288 ((8 does per C1 and C2 group, respectively) × 6 breeding cycles). C1: group housing, C2: semi-group housing. *T*: period effect, *S*: System effect. Different letters on the same rows ^(a,b)^ mean different *p* values.* = *p* ≤ 0.05, ** = *p* < 0.01, n.s. = *p* > 0.05.

**Table 4 animals-09-00528-t004:** Effect of housing system on reproductive performance of does.

Housing System	Unit	C	C1	C2	*p*-Value	MSE
Sexual receptivity	%	85.2 ^c^	72.6 ^a^	79.2 ^b^	*	4.52 ^†^
Fertility rate	%	82.8 ^b^	69.3 ^a^	76.2 ^b^	*	3.41 ^†^
Kindling rate	%	78.3 ^c^	60.2 ^a^	69.0 ^b^	*	4.08 ^†^
Embryo mortality	%	4.5 ^a^	9.1 ^b^	6.2 ^a,b^	*	0.50
Doe weight at kindling	g	3750	3455	3540	n.s.	565
Doe weight at weaning	g	4220	3860	3985	n.s.	385
Estimated depot fat at AI	g	53 ^b^	32 ^a^	40 ^a^	**	1.58
Born alive	*n*	8.90 ^b^	7.50 ^a^	7.95 ^a^	**	0.32
Weaned pups	*n*	7.85 ^c^	6.91 ^a^	7.20 ^b^	*	0.21
Weight at weaning	g/pup	585	570	565	n.s.	39.2
Pre-weaning mortality	%	5.5 ^a^	8.0 ^c^	7.2 ^b^	*	0.75 ^†^

*n* = 288 ((16 does per experimental group) ×6 breeding cycles). C: conventional, C1: group, C2: semi group. Different letters on the same rows ^(a–c)^ mean different *p* values.* = *p* ≤ 0.05, ** = *p* < 0.01, n.s. = *p* > 0.05. ^†^: χ^2^ value.

**Table 5 animals-09-00528-t005:** Effect of housing system on indexes of global productivity.

Housing System	Unit	C	C1	C2	*p*-Value	MSE
Rabbits sold/year/doe		35.5 ^c^	20.4 ^a^	25.6 ^b^	*	2.68
Live weight sold/year/doe	kg	81.6 ^c^	46.6 ^a^	60.7 ^b^	***	4.29
Production losses	kg	42.4 ^a^	73.5 ^b^	48.5 ^a^	**	3.58
Kindling interval	day	75.2 ^a^	94.3 ^b^	82.1 ^a,b^	*	5.82
Kindling/year/doe	n.	4.80	3.78	4.35	n.s.	0.58
Annual doe replacement	%	75.0 ^a^	112.0 ^b^	87.5 ^a^	**	6.57 ^†^

*n* = 288 ((16 does per group, respectively) ×6 breeding cycles). C: conventional, C1: group, C2: semi group. Different letters on the same rows ^(a–c)^ mean different *p* values.* = *p* ≤ 0.05, ** = *p* < 0.01, *** = *p* < 0.001. n.s. = *p* > 0.05.^†^: χ^2^ value.

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
