# Peer review of "Housing Rabbit Does in a Combi System with Removable Walls: Effect on Behaviour and Reproductive Performance"

_animals, 2019, doi:10.3390/ani9080528_

Round 1
Reviewer 1 Report
The present study assesses welfare and productive performance of domestic does under different group-housing conditions. Although the trade-offs between the welfare detriments of isolation versus aggression due to group housing in does are highly relevant, the study has fundamental issues.
The methodology needs extensive clarification. If I understand correctly, does that were non-pregnant were replaced with pregnant does. In C1 and C2 there were 16 does distributed in 4 production units, giving 4 experimental units per treatment of group housing (C1 and C2). Did the replacement does belong to the 16 animals? How many non-pregnant does were replaced in each system? Different replacement and regrouping is a confounding on the study, and without clarification and control in the models, no strong conclusion could be drawn from this data. In addition, authors state that 8 does in each system (C1 and C2) were randomly selected for behavioural observation, which if I understood correctly would correspond to two does per production unit. Hence, doe nested within production unit needs to be included in the models as a random effect.
Behavioural measurements have questionable reliability since no ethogram with description of behaviors or inter-observer reliability was presented. These issues limit the validity of the conclusions.
In addition, some parts of the introduction were textually copied from published texts that without quotation marks, this is considered plagiarism.
Lines 61-62: “The leporaria were the 62 origin of the warrens or game parks that subsequently developed in the Middle Ages [2]” From Lebas, et al (1997).
Author Response
The methodology needs extensive clarification. If I understand correctly, does that were non-pregnant were replaced with pregnant does. In C1 and C2 there were 16 does distributed in 4 production units, giving 4 experimental units per treatment of group housing (C1 and C2). Did the replacement does belong to the 16 animals? How many non-pregnant does were replaced in each system? Different replacement and regrouping is a confounding on the study, and without clarification and control in the models, no strong conclusion could be drawn from this data. In addition, authors state that 8 does in each system (C1 and C2) were randomly selected for behavioural observation, which if I understood correctly would correspond to two does per production unit. Hence, doe nested within production unit needs to be included in the models as a random effect.
The purpose of the study was to have more space per doe (colony =4 cages) without changing the dimensions of single cage. In this way the higher cost of equipment isdiminished.
As explained in several parts of the paper, the experiment has been plannedin view ofa possible real application. According to this view, it is not conceivablethat not productive does (non pregnant) rest in the colony because the production level of the colony cage would be very low. Accordingly, it is obligatory to have “redundant” does for replacing non-pregnant does at each AI cycle. So, as written in the text, after abdominal palpation, executed at 12 d post AI, the non-pregnant does were replaced with pregnant ones hosted in the upper level of the cage (C2) or in supplemental single cages (C and C1). Considering the average fertility rate (82.8, 69.3 and 76.2%) of the three experimental groups, about 1 doe/cycle was replaced because not pregnant. Clearly as already addressed the number of replaced does is lightly different depending on the fertility rate at AI.
Behavioural measurements have questionable reliability since no ethogram with description of behaviors or inter-observer reliability was presented. These issues limit the validity of the conclusions.
Ethogram with categories of behavior patterns of does was added in material and methods. As reported in line 203, the behavior observation was done by two experts trained to work together, this means that the error between the two operator’s observations was practically avoided. We have only specified the most difficult to detect behavior in lines 187-192.
In addition, some parts of the introduction were textually copied from published texts that without quotation marks, this is considered plagiarism.
Lines 61-62: “The leporaria were the 62 origin of the warrens or game parks that subsequently developed in the Middle Ages [2]” From Lebas, et al (1997).
We modified the introduction as followed “The origin of the Leporia dates back to the Middle Ages following the delimitation of areas destined for hunting”.

Reviewer 2 Report
General comments: This study evaluated welfare and reproductive parameters of does housed for ~1 year eithr in isolated, grouped or periodically grouped cages. No additional space/cage resources were provided to any animal in any housing system. More locomotion and some positive social behaviors were noted in does housed with partial social contact; however, the highest reproductive index and best body condition was noted in breeding does that were single housed.
Specific comments:
Style: Minor grammar errors throughout.
2.2 Experimental design:
pg 4, line 134 – is this a weakness of the study – could does have been better distracted with a combination of solid walls and enrichment? The all-mesh design does not provide for any visual barriers and there is little for animals to do all day in an otherwise barren cage, regardless of compartmentalization. Please address. Also, having more space to move around is supposed to be an intended benefit of enlarged cages – housing at the same density as conventional cages does not address this issue at all – another weakness of the study/cage design.
Pg 4, line 136 – how was randomization conducted, e.g., random number generator? Please include the age (weeks) of does at study start. What was the source of the does? Please explain how controlled lactation was conducted.
Pg 6, line 178 – please indicate the source (supplier) of the feed. Was the diet unchanged regardless of where the does were in their production cycle? Were any parasiticides or antimicrobial agents added to the diet? Please include.
Pg 6, line 183 – how were does selected for behavioural observations? Were the same 8 does/group followed for the entire year or did they change (e.g., for reasons of poor productivity, disease, injury, etc)? What was used to mark the does for observation? Were the selected does blocked by cage type and number?
Pg 6, line 203 – what was the degree of concordance for the 2 behaviour scorers?
Pg 6, line 207 - What was the rationale underlying waiting only 5 min for animal habituation before scoring?
Pg 6, line 209 – How long was each animal observed in the single vs the 2 types of social cages?
Pg 8, line 243 – statistics section - why are some of the sentences bolded?
Line 316 (note – page numbering is lost after Table 1) – you’ve only given the location of the injury and the proportion injured. Please also include a table indicating and comparing the severity of injuries in does in C1 vs C2 systems. Were any does or kits removed from the study because of injuries?
Line 341 – when you indicate ‘different EE strategies were used’, are you referring to different studies by different authors? Previously, you indicate that no other enrichments were used and the cages were barren.
Line 351 – this is a weakness of this paper – more space and visual barriers should be considered
Figure 1 – there are 2 images – please label ‘A’ and ’B’ and add information to enhance clarity – is one the front view and the other the top view? Also, in the first image, the dimension above indicates 150 cm but the smaller arrow in the actual figure is labeled as 150 cm and the bigger line above it is unlabeled. Please ensure that each drawn line is labeled with a dimension.
Figure 3 & 4 – please change ‘productive protocol’ in first sentence of figure headers to ‘rabbit production protocol’.
Table 2 – does this represent the average # of injuries across all reproduction cycles or only 1 cycle? Please clarify. If the former, what was the variation in injuries seen across cycles (i.e., please present as mean +/- SD or SE).
Author Response
Specific comments:
Style: Minor grammar errors throughout.
Ok. The manuscript was subjected to linguistic revision (see language certificate).
2.2 Experimental design:
pg 4, line 134 – is this a weakness of the study – could does have been better distracted with a combination of solid walls and enrichment?
Yes, it is known that some enrichments have a positive effect on animal welfare. However, the aim of the study was to compare the productive performance and welfare of animals in three different typologies of cages. It was difficult to add the same enrichment in these cages because they have many structural differences.
The all-mesh design does not provide for any visual barriers and there is little for animals to do all day in an otherwise barren cage, regardless of compartmentalization. Please address.
OK. With the barren cage, we meant referred that the cages did not have any enrichment inside. The cages used in this study are the same used in commercial farms to breed rabbits. Then the cages did not present any visual barriers for the animals. However, the isolation of animals with solid wall is retained as an obstacle to social interactions from neighboring does.
Also, having more space to move around is supposed to be an intended benefit of enlarged cages – housing at the same density as conventional cages does not address this issue at all – another weakness of the study/cage design.
We agree, but unfortunately, in commercial condition (breeding farm), the enlargement of available space (mainly in term of surface area/rabbit) greatly increases the production costs. The effort is to have more space for the four does for animal movement also if the animal density was the same.
Pg 4, line 136 – how was randomization conducted, e.g., random number generator?
If the reviewer referred to the does choose, Yes, it is. We chose the does without take into consideration any productive/reproductive traits because the female rabbits were homogenous nulliparous New Zealand White rabbit does of the same age (120 days). Regarding the replacement of does the non-pregnant does were randomly replaced with pregnant ones hosted in the upper level of the cage (C2) or in supplemental single cages (C and C1).
Please include the age (weeks) of does at study start. What was the source of the does?
The does at four months of age were purchased by the Italian National Rabbit Genetic Reference Center (Foggia, Italy). Please, see line 144-147.
Please explain how controlled lactation was conducted.
Done in text. Please, see line 153-154
Pg 6, line 178 – please indicate the source (supplier) of the feed. Was the diet unchanged regardless of where the does were in their production cycle? Were any parasiticides or antimicrobial agents added to the diet? Please include.
The characteristics of the diet used were reported in lines 178-180. The diet was the same during the reproductive cycle and came from a commercial feed mill.
Pg 6, line 183 – how were does selected for behavioral observations? Were the same 8 does/group followed for the entire year or did they change (e.g., for reasons of poor productivity, disease, injury, etc)? What was used to mark the does for observation? Were the selected does blocked by cage type and number?
The eight does/group changed in a relation to the reproductive cycle. In each productive cycle, eight does/group were randomly chosen and marked with a spray of different colors as reported in lines 184-185. The does were marked for the individual identification (added in the text).
Pg 6, line 203 – what was the degree of concordance for the 2 behaviour scorers? If you mean experts and trained to work together
We used the “focal animal sampling method” or continuous sampling method, which consist in: watch 1 animal and record all of its activities for a predetermined period of time; the observers were trained and the variability between observer was almost zero.
Pg 6, line 207 - What was the rationale underlying waiting only 5 min for animal habituation before scoring?
The trial was conducted in the experimental farm of the University of Perugia where every day (two times a day)an observer, already present in the rabbit shed, made the examination. For these reasons, animals were adapted to the human’s presence that is in any case very discreet. In our experience, five minutes is enough time for permitting the animals to adapt to the observer presence without changing their behavior.
Pg 6, line 209 – How long was each animal observed in the single vs the 2 types of social cages?
Rabbit does never changed the housing system to which were initially assigned. In the text, the observations times for each doe of the two colony and standard cages are reported at lines 231-235 (3’ for standard cage does whereas 6’ for colony ones).
Pg 8, line 243 – statistics section - why are some of the sentences bolded?
Probably it was an error of formatting. We changed them.
Line 316 (note – page numbering is lost after Table 1) –
The mistake does not depend on us but on the journal template.
you’ve only given the location of the injury and the proportion injured. Please also include a table indicating and comparing the severity of injuries in does in C1 vs C2 systems. Were any does or kits removed from the study because of injuries?
The reviewer note is correct; however, considering the amount of data and the difficulty to clearly discuss them we preferred to regroup the injures scores as % of animals mainly belonging to score 2. We better explained it on M&M and results (Please, see line 336 342).
Line 341 – when you indicate ‘different EE strategies were used’, are you referring to different studies by different authors? Previously, you indicate that no other enrichments were used and the cages were barren.
Yes, it was referring to other studies reported in the references list [27,29,30,31,32,33,7]
Line 351 – this is a weakness of this paper – more space and visual barriers should be considered. We agree, but we wanted to study a possible compromise, that give to rabbit does social behavior, but do not increase much the cost of production for farmer.In all EU, only a very low percentage of rabbit farm changed the rearing system (changing the cage for does or fattening) due to the excessive costs of equipment. Moreover, if visual barriers would be present, the isolation of animals would be more pronounced, which is an opposite situation respect to the aim (the group housing) of the research.
Figure 1 – there are 2 images – please label ‘A’ and ’B’ and add information to enhance clarity – is one the front view and the other the top view? Also, in the first image, the dimension above indicates 150 cm but the smaller arrow in the actual figure is labeled as 150 cm and the bigger line above it is unlabeled. Please ensure that each drawn line is labeled with a dimension.
Ok, done. We also add two images of the colony cages.
Figure 3& 4 – please change ‘productive protocol’ in first sentence of figure headers to ‘rabbit production protocol’.
Ok, done.
Table 2 – does this represent the average # of injuries across all reproduction cycles or only 1 cycle? Please clarify. If the former, what was the variation in injuries seen across cycles (i.e., please present as mean +/- SD or SE).
As reported under table 3, the value represents the average of injuries across all reproduction cycles. So, the variation in injures is relative to the six reproduction cycles.

Reviewer 3 Report
Regarding to manuscript ID animals-522316 entitle “Housing rabbit does in a combi system with removable walls: Effect on behaviour and reproductive performance”, the authors aimed to evaluate the effect of cage type (in individual conventional cages or in different colony cages: simple colony cages (does in the group for 100% of the reproductive cycle) or combi colony cages (individual and in group caging) and housing condition (total isolation vs. group-housing vs. semi-group-housing) on welfare and reproductive performance of rabbit does. The experiment is well designed and lasted for one year (6 reproductive cycles). This research provides some useful information of improving rabbit raising system to balance both requirements of animal welfare and production efficiency. I recommend acceptance of this manuscript after revision.
Comments for revision:
1. Rabbits were raised in an environmental controlled house with a broad daily temperature range from 15 to 28°C. Were the 3 groups of rabbits raised in the same house or in houses with well-controlled uniform environment? Since the environment sometimes have bigger impact on animal’ behaviours than the raising system, so please provide synchronous local environment indexes of 3 groups of rabbits.
2. Type and frequency of feeding also affect rabbit behavior. Please add the feeding information in the manuscript. Were the rabbits feed by manual for several times per day or feed automatically with ad libitum?
3. It’s hard to understood how the rabbit behaviors were recorded in duplicate on site by two trained operators (Line 209-211). If I understand correctly, had the behaviours been recorded with videos in duplicate? If so, what is the time sequence and interval in recording the 3 groups of rabbit behaviors ( 8×3 rabbits) ? Or were those recorded at same time?
4. Please add the time unit in Table 1. Is it “min”?
5. In Line 227-229 “fertility rate was established by abdominal palpation at 12 days after AI. Late embryo mortality was estimated as the difference between fertility and kindling rate.” I suggest change the “embryo mortality” to “fetal mortality” or “fetal loss rate”.
6. The description for “Production losses” (in Line 233-235) is not clear. How is “Production losses” calculated? It’s hard to evaluate whether it is accurate or not.
Author Response
Comments for revision:
1. Rabbits were raised in an environmental controlled house with a broad daily temperature range from 15 to 28°C. Were the 3 groups of rabbits raised in the same house or in houses with well-controlled uniform environment? Since the environment sometimes have bigger impact on animal’ behaviours than the raising system, so please provide synchronous local environment indexes of 3 groups of rabbits.
Ok, we clarified it in M& M.
2. Type and frequency of feeding also affect rabbit behavior. Please add the feeding information in the manuscript. Were the rabbits feed by manual for several times per day or feed automatically with ad libitum?
Ok, done. Please see line 190-193
3. It’s hard to understood how the rabbit behaviors were recorded in duplicate on site by two trained operators (Line 209-211). If I understand correctly, had the behaviours been recorded with videos in duplicate? If so, what is the time sequence and interval in recording the 3 groups of arbbit behaviors ( 8×3 rabbits) ? Or were those recorded at same time?
Behaviors were gathered by two researchers directly through visual observation for each experimental does (see before, answers to REV 1 and REV 2).
4. Please add the time unit in Table 1. Is it “min”?
Done, please see M&M and Table 2.
5. In Line 227-229 “fertility rate was established by abdominal palpation at 12 days after AI. Late embryo mortality was estimated as the difference between fertility and kindling rate.” I suggest change the “embryo mortality” to “fetal mortality” or “fetal loss rate”.
Ok, done.
6. The description for “Production losses” (in Line 233-235) is not clear. How is “Production losses” calculated? It’s hard to evaluate whether it is accurate or not.
We mean with production losses the difference (in kg) between the maximal theoretical production (considering fertility rate = 100, the mortality of the young rabbits = 0 and kindling interval = 60), and the real one. Anyway, we clarified it on the text (see line 250-254)

Round 2
Reviewer 1 Report
1. It is understandable that from an applied purpose, the replacement of does was necessary. However, the impact of different number of replacements across treatments should be controlled for in the analysis, or should be acknowledged in the discussion. How does re-grouping alters behaviour and order response variables?
2. The Ethogram should include detailed and unambiguous definitions of the behaviours. Should the experiment be replicated, the behaviours measured need to be clearly defined.
3. Although training indeed reduces error between observers, a measure of the inter-observer consistency should be presented. I understand this may not be possible, however, it should still be acknowledged.
Author Response
Dear Reviewer
Thank you for the opportunity to revise our manuscript “Housing rabbit does in a combi system with removable walls: Effect on behaviour and reproductive performance”.
We appreciate your careful and constructive suggestions.
It is our belief, after making the suggested edits, that the manuscript is substantially improved, and our nex studies too.
Following this letter are our responses to comments, including our Microsoft Word with "Track Changes" function on, so that changes are easily visible.
The revision has been developed in consultation with all coauthors, and each author has given approval to the final form of this revision.
Thank you for your suggestions.
Cecilia Mugnai

This manuscript is a resubmission of an earlier submission. The following is a list of the peer review reports and author responses from that submission.
Round 1
Reviewer 1 Report
I would have liked to see more of a discussion on rabbit welfare as opposed to primarily performance. Other than that, the paper seems fine to me, although it needs editing for English language.
Reviewer 2 Report
Reviewers comments: Housing rabbit does in a combi system with removable walls: Effect on behaviour and reproductive performance
Simple summary: “60% timesheet” doesn’t really make much sense in a lay summary, assuming that ‘timesheet’ is a term you’re used to, but others won’t be. Same for the term ‘peri-partum’, this is fine for biologists but not for a lay audience. Same for “global efficiency indexes”. This lay summary really isn’t ‘lay’ and fails to effectively communicate the key points of the study.
The English language needs some improvement, particularly with respect to grammar. Examples of this in the Introduction are: in line 2 where it says, “independently on the larger area for moving and social contact”, I assume you meant the word ‘on’ to be ‘of’; also line 55-56 “the reproductive performance are lower” should ideally read, ‘the reproductive performance of group housed rabbits were lower’. Line 57 “single and group housing is studied” should read ‘single and group housing has been studied’. Additionally, line 67, “and should be modify into a more efficient way”, isn’t good English, grammatically speaking. In line 71, I don’t really know what this means, “without any external need”. Line 74, “At our knowledge” should be “To our knowledge”, and line 75, “consisting in does” should be ‘consisting of does’.
Other than the language issues, the Introduction does not in my opinion provide sufficient background and is highly imbalanced. I am in favour of brief introductions, but this is far too brief. There is no mention that does in the wild spend just minutes per day with their kits, and thus the artificial nature of isolating a doe in with her kits in breeding systems. There is also no mention of the natural hierarchies formed by group living rabbits, and thus why ‘regrouping’ is a disruption that would be expected to lead to aggression in the short-term (this was mentioned later in the discussion). The authors say that continuous group housing contradicts welfare standards, and cite example of reduced welfare in such housing, without actually acknowledging the need for rabbits to have access to social companionship, which of course means that single housing also compromises welfare, just in different ways. This introduction is not balanced, it provides a setup of background that only focuses on the point of view of the producer (i.e. reduced reproductive productivity) and fails to focus more broadly on the various issues surrounding the welfare of breeding does.
It also fails to set the scene for the purpose of the study regarding the type of rabbits being studied: are these cage systems traditionally used for food farming or for laboratory breeding colony production? What production system are you emulating?
Figure 2 has obviously taken time to put together accurately but it is far too difficult to understand from this figure what the actual unit looked like and what space was available to the rabbits in the different phases. The authors say, “In the colony cages (both C1 and C2), rabbit does were housed at the same density of the conventional system (C)”, does this mean they had 2,7000px2 of space per rabbit? Does this include the space in the nest-boxes?
If the below dimensions are correct, then the singly caged animals do not have anywhere near enough space to be physically healthy in terms of their ability to stretch out and exercise, both of which are fundamental to rabbits maintaining skeletal health.
Colony cage dimensions: 76W x 150L x 60H cm (floor area 11,400cm2)
Combi colony cage: 130W x 158L x 46H cm (floor area 20,540cm2)
Conventional single cage: 38W x 60L x 34H cm (floor area 2,80cm2)
All of these cages were below the minimum height required (75cm) to allow an average sized rabbit to stand and stretch; a natural behaviour that is instantly being deprived in all of the housing sizes used in this study. Furthermore, New Zealand Whites are categorised as large breeds, so their minimum space requirements (space to stretch out, hop at least three body lengths and to allow for exercise) are not likely to be being met by any of these cage designs even if house alone in the biggest. This isn't acknowledged at all, anywhere in the paper, but if you want it to be about the welfare of the rabbits then it must be acknowledged.
Am I to understand from section ‘2.3 Diets’ that these rabbits aren’t provided with hay? That they instead get all their energy requirements in short high-quality meals? If yes, then this is also a significant welfare concern that will be exacerbating the poor conditions these rabbits are kept in (in addition to space restrictions). It is therefore, no wonder that the does fight each other when they’re cramped together, unable to exercise, and left with nothing to do all their waking hours when they should, biologically speaking, be eating low quality forage for the majority of their waking hours. It is possible that the stereotypic behaviour of the singly housed does could even be reduced if they were provided with hay which was supplemented with pellets.
Were these empty unenriched cages? No information is provided regarding any provision of bedding material, places to hide or things to chew, so I assume they are barren – this should be stated explicitly.
Lines 170-173: More detail needed here. I see you used focal sampling, but did you record using continuous-all occurrence sampling for your focal animal? I would assume so, but this has not been made clear. And how long did you observe a focal animal for before moving on to the next one? You say in line 176-7: “observation was extended by further 3 min” but did not say what the observation period was in the first place….I see this is given in lines 179-81, please move this up higher so the reader has this information first.
Table 1. has not displayed properly in the PDF. Also ‘MS’, ‘T’, ‘S’ and ‘TXS’ have not been defined.
Line 306: sorry but this makes no sense, “avoided the exchange of nest negative for the litter’
Line 317-319: You say, “Possibly, the “calmer” (more protected) environment also affected the energy equilibrium of does and colony-caged does consumed more energy for interacting with the other does.” This puts a somewhat biased spin on the situation, it could also be argued that the environment of the single does was ‘deprived’ as opposed to ‘calm’, and yes, having greater space availability to move and interact with other rabbits will use up more energy, but this could be overcome by simply feeding them more, as the current guidelines are likely based upon data from energy consumption of singly housed, space restricted does.
Line 325: you say, “as confirmed by the percentage of does injured, culled and replaced”, yet (forgive me if I just missed this) culling and replacement data has not been presented thus far. This is also backed up in line 359 where it says, “the high percentage of severely injured does” but this figure has not been given.
Line 342: “they lived in stressful conditions” is not a good way to say this. It would be more accurate to say, ‘they experienced social stress’. All of the rabbits are living in stressful, unnatural laboratory-based conditions.
Line 359-61: This is a terribly flawed conclusion: “Because of high stress increased morbidity and mortality, C1 does had poor animal welfare and body condition (severe body injuries and low fat depots) and lower productivity”.
Ok, yes, they definitely had poorer physical condition, and increased social stress, but that does not automatically equate to “poor animal welfare”. The lower fat depots are likely because the diet is based on data for energy use of singly housed does, and that can be easily adjusted, so that isn’t a factor relevant to whether the cage system could work. The body injuries and aggression are a bad thing, of course, but as you say yourself that is because of re-grouping not the fact that they were social per say, so if regrouping could be avoided this would be ok. Other injuries and aggression were from rabbits entering other nests, which you also hint could be prevented by nest training although this section is not adequately described if the reader has not read your previous study. However, you also provide data that shows the rabbits are behaviourally much better-off, with significantly lower stereotypies and more time spent expressing natural behaviour. In fact, their behaviour alone would indicate improved welfare, whilst their body condition indicates reduced physical wellbeing, so your results are contradictory for overall welfare. ‘Welfare’ does not equal ‘physical health’, despite their inter-relatedness. The results are quite clear from a production point-of-view, but are not clear at all from a welfare stand point. Overall, knowing the welfare needs of rabbits, I would say that none of these setups is meeting their needs, and all are depriving them of their Five Freedoms in one way or another. This, along with the broader welfare considerations of whatever production system you are describing here is not discussed at all, which has greatly disappointed me.